# Ordinary and Extraordinary Complex Refractive Indices Extraction of a Mylar Film by Transmission Spectrophotometry

**DOI:** 10.3390/polym14091805

**Published:** 2022-04-28

**Authors:** Yassine Makhlouka, Fadhel Sanaâ, Mohamed Gharbia

**Affiliations:** 1Laboratoire de Physique des Liquides et d’Optique Non Linéaire, Département de Physique, Faculté des Sciences de Tunis, Université de Tunis El Manar, Tunis 1068, Tunisia; fadhelsanaa@ipeib.u-carthage.tn (F.S.); mgharbia2@yahoo.com (M.G.); 2Département de Physique-Chimie, Bizerte Preparatory College for Engineering Studies (IPEIB), Carthage University, Amilcar 1054, Tunisia; 3Laboratoire de Nanomatériaux et des Systèmes pour les Énergies Renouvelables (LANSER), Centre de Recherches et des Technologies de l’Energie, Technopole Borj-Cedria, Université Tunis El Manar, Tunis 1068, Tunisia

**Keywords:** transmittance spectrophotometry, complex refractive index, extinction coefficient, birefringence, diattenuation, dichroism, Cauchy model, dispersion

## Abstract

A new spectrophotometric technique for the determination of both ordinary and extraordinary complex refractive indices (CRIs) of a stretched polyethylene terephthalate (Mylar) film is proposed. The sample was placed between two identical polarizers, and the transmission spectra of two different configurations (incident polarization parallel and perpendicular to the Mylar film optical axis) were recorded. Ordinary and extraordinary complex refractive indices are then extracted by fitting the experimental spectra with a theoretical model that we had elaborated in advance. A new formula for transmittance dispersion, based on the Fresnel’s coefficients formalism and using the Cauchy model, was derived to describe *n* and *κ* wavelength dependence. The suggested theoretical model succeeded in reproducing the Mylar transmission spectra across the entire visible spectral range (400, 750 nm) for both configurations, and the retrieved dispersion curves of the refractive indices, extinction coefficients, and the birefringence are comparable to results found in the literature. The proposed method is fast, straightforward, easy to set up, and cost-effective. It proved to be an excellent alternative to more conventional methods such as spectroscopic ellipsometry.

## 1. Introduction

Polymer films such as biaxially stretched polyethylene terephthalate (PET), commercially known as Mylar, have been subjected to extensive studies, owing to their various applications in optoelectronics, such as organic light-emitting devices (OLED) and displays. In addition to their lightness, flexibility, and low cost, their optical properties can be precisely tuned to satisfy specific display requirements. Mylar transparent films are typically used in liquid crystal displays (LCD) as optical spacers [1,2] or as a retardation plate to compensate the birefringence of the liquid crystal layer and obtain a wide viewing angle [1]. It is crucial to measure the different optical properties of Mylar films in order to optimize the display’s characteristics and obtain high performances, such as high contrast ratio and low color shift. Moreover, knowing the optical properties of these materials offers a better insight into the effect of uniaxial or biaxial stress, applied during the manufacturing process, on the orientation of polymeric molecular chains and the morphologies induced by various treatment techniques and conditions [3,4]. One of the most useful optical properties of such materials is the complex refractive index (CRI) denoted by $\overset{ˇ}{n}$: a quantity that characterizes interactions between the polymer film sample and incident light. Its real part n, called refractive index, informs about the change of incident light direction (refraction), and its imaginary part κ, called extinction coefficient, describes the light intensity loss due to absorption and scattering. Furthermore, stretched polymer films exhibit uniaxial anisotropy in their optical properties; therefore, they have two different complex refractive indices corresponding to two principal polarization states: extraordinary (ECRI) and ordinary (OCRI) indices. The difference in the real parts of (ECRI) and (OCRI) gives rise to the birefringence, and the discrepancy in their imaginary parts leads to the determination of the diattenuation. It has been reported [5,6] that birefringence in linear retarders, such as Mylar films, is generally accompanied by some amount of diattenuation, due to the difference between the extraordinary and ordinary absorption coefficients (dichroism) and to a discrepancy in the Fresnel’s coefficient of reflection and transmission at the interfaces. Knowing the extraordinary and ordinary CRIs of Mylar films, for any given wavelength in the visible range, is essential to achieve a more precise and efficient birefringence compensation in (LCD) devices. On the other hand, the measurement of the CRI will enable the calculation of the dielectric function tensor in the corresponding spectral region, which gives access to a large number of fundamental physical properties.

Several experimental works have suggested various techniques for the determination of the CRI spectral dispersion of stretched PET films in various regions of the electromagnetic spectrum. Loewenstein and Smith [7] used a Fourier spectroscopic interferometry to analyze both reflection and transmission channeled spectra of a Mylar film in the far-infrared range. The real part of the CRI was extracted from the fringes’ positions and the imaginary part from the fringes’ intensities. Another approach suggested by Sergides et al. [8] and based on mid-infrared spectroscopic reflectometry conducted on a PET sample consisted of measuring the ratio of p-polarized light reflectance to the s-polarized light reflectance for two different incident angles. Then, n and κ were calculated, for each wavelength, by matching the measured data with a constructed theoretical model. Ouchi et al. [9] recorded the reflection spectra of uniaxially drawn PET films in the far-ultraviolet and used the Kramers–Kronig formula to calculate the absorption spectra. The refractive index and the absorption coefficient were calculated in a next step, using Fresnel’s formulae. Seres et al. [10] measured the transmittance of various commercial type-D Mylar sheets with different thicknesses and for a discrete number of visible wavelengths. Absorption and reflection coefficients were then obtained by fitting the thickness dependence function with Beer’s transmittance formula. Among all proposed techniques for semi-transparent polymer films optical characterization, ellipsometry remains by far the most successful one. It owes its usefulness to its ability to extract both sample thickness and CRI. Various ellipsometric methods have been proposed in the literature; among them, three are worth mentioning. In the work of Zhang et al. [11], the spectroscopic ellipsometry technique combined with the ray-tracing method was employed to measure the complex refractive index of amorphous PET samples in the broadband 0.4–2 µm. Losurdo et al. [12] used a variable angle spectroscopic ellipsometry to retrieve the CRI dispersion curves of conjugated polymeric films in the 1.5 eV–5 eV range. Using the Lorentzian dispersion function, they succeeded in modeling the sample spectral dependence of in-plane and out-of-plane CRIs and estimate both its birefringence and dichroism. Another ellipsometric technique proposed by Kostruba et al. [13] allowed the independent determination of strongly correlated refractive index and thickness of transparent ultrathin films of thicknesses ranging between 1 nm to 20 nm.

Despite these previous works and employed methods, we noticed a lack of data regarding extraordinary and ordinary components of Mylar CRI in the visible range of the spectrum. Light attenuation through polymer films is generally attributed to absorption induced by infrared molecular vibrations and electronic transitions in the short wavelength ultraviolet region [14]. Thus, their extinction coefficients are usually smaller in the visible to near-infrared bands and larger in the mid-infrared to the far-infrared band. Transmission and reflection methods described in the references [7,8] are particularly applicable in the infrared region and cannot provide a direct accurate measurement of the extinction coefficient in the visible band. Moreover, these methods are generally used in conjunction with a Kramers–Kronig (KK)-based calculation [15,16,17,18] which requires, in principle, information over the whole spectrum, while measurements provided by these methods are unavoidably finite. Many strategies were proposed to overcome this serious practical limitation, such as the use of Roessler’s approximation [19], extending the integration to infinity by an a priori choice of asymptotic behavior outside the measured data range [20], or through the modified Multiply Subtractive Kramers–Kronig formula [21], but these solutions are either heuristic [19,20] and could lead to erroneous results or require the knowledge of extra discrete values of CRI to be obtained from independent measurement [21]. In this work, we report a spectrophotometric method to obtain the CRI dispersion function of a Mylar film from a single measurement of its transmission spectrum in the visible band. Recorded experimental transmittances are then fitted to a theoretical model based on Fresnel’s equations [22,23,24] and the Cauchy dispersion formula [15]. Ordinary and extraordinary transmittance spectra are recorded for two different configurations where incident light is linearly polarized parallel and perpendicularly to the sample’s principal axis, allowing a simultaneous determination of the OCRI and ECRI. During this study, the Mylar film will be treated, in a first approximation, as a plane-parallel plate of a uniaxial medium with its optical axis parallel to its surface. Compared to the previously described techniques, this method is simple and easy to set up, as it only requires a spectrophotometer and polarizers, and provides significant savings in work and time.

## 2. Theoretical Background

### 2.1. Optical Properties of Uniaxial Materials

The optical properties of transparent linear anisotropic media depend on the direction of propagation and polarization of incident light. In other words, the refractive index describing the dielectric properties of the material is a tensor quantity for which there always exists a system of eigenvectors $\langle u_{x},\;u_{y},u_{z}\rangle$, called the system of principal axes, in which this tensor is diagonal with three different eigenvalues corresponding to the principal refractive indices $n_{x,y,z}$. In the particular situation of uniaxial material, one of the three principal axes (*x* for instance) is a symmetry axis and defines a privileged direction called the optical axis. As a result, *y* and *z* axes are physically equivalent with $n_{y}=n_{z}=n_{o}$ and $n_{x}=n_{e}$. Here $n_{o}$ and $n_{e}$ are called ordinary and extraordinary indices, respectively, and $\Delta n=n_{e}-n_{o}$ is known as birefringence [25,26,27]. This property introduces a velocity difference between light polarized along the two different principal axes of the uniaxial material. Note that in the case of a positive uniaxial medium where $n_{e}>n_{o}$, such as stretched PET films exhibiting molecular orientation [28], the direction of the optical axis is often referred to as the slow axis. If we consider a linearly polarized light ray with a direction defined by the wave vector k along the *z*-axis propagating at normal incidence through a uniaxial plane-parallel plate with its optical axis (*x*-axis) lying in the plate plane, the incident and transmitted lights are defined by two transverse electric fields as follows:(1)$$E_{incident}=E=E_{0}{\left.e^{i\left(\omega t-kz\right)}\right|}_{z=0}=E_{0}e^{i\omega t}=\left(E_{0x}u_{x}+E_{0y}u_{y}\right)e^{i\omega t}$$
(2)$$E_{transmitted}=E^{\prime }=E_{0}^{\prime }{\left.e^{i\left(\omega t-kz\right)}\right|}_{z=L}=E_{0}^{\prime }e^{i\left(\omega t-kL\right)}=\left(E_{0x}^{\prime }u_{x}+E_{0y}^{\prime }u_{y}\right)e^{i\left(\omega t-\phi \right)}$$
where L is the plate thickness and $\phi =\frac{2\pi nL}{\lambda }$ is the introduced phase at the exit interface due to propagation. In the special case of uniaxial waveplates [27,29], the impinging light is decomposed into an e-ray component polarized parallel to the plate optical axis and traveling with the phase velocity $v_{e}=\frac{c}{n_{e}}$ and an o-ray vibrating along the *y*-axis with a phase velocity $v_{o}=\frac{c}{n_{o}}$. Each polarization state is defined by a different electric field as follows:(3)$$\left\{\begin{array}{c} E{{}^{\prime }}_{x}=E{{}^{\prime }}_{e}=E_{0x}^{\prime }e^{i\left(\omega t-\phi e\right)\;}u_{x}\\ E{{}^{\prime }}_{y}=E{{}^{\prime }}_{o}=E_{0y}^{\prime }e^{i\left(\omega t-\phi o\right)\;}u_{y} \end{array}\right.$$

In addition, if the medium is non-lossy then, $E_{0x}^{\prime }=E_{0x}$ and $E_{0y}^{\prime }=E_{0y}$, which gives:(4)$$\left\{\begin{array}{c} E{{}^{\prime }}_{x}=E_{x}e^{-i\phi e}\\ E{{}^{\prime }}_{y}=E_{y}e^{-i\phi o} \end{array}\right.$$

Consequently, the propagation of the light wave through the uniaxial waveplate induces a phase retardation Δ between the components of the electric field defined by:(5)$$\Delta =\phi e-\phi o=\frac{2\pi \Delta nL}{\lambda }$$

Using the Jones calculus formalism [30], incident and transmitted light waves can be modeled by two Jones vectors $\left[E\right]$ and $\left[E^{\prime }\right]$, and the linear retardation effect of the uniaxial plane-parallel plate is described using a 2×2 Jones matrix $\left[J\right]$ defined by:$$\left[E\right]=\left[\begin{array}{c} E_{x}\\ E_{y} \end{array}\right]\;;\;\left[E^{\prime }\right]=\left[\begin{array}{c} {E^{\prime }}_{x}\\ {E^{\prime }}_{y} \end{array}\right];\;\left[J\right]=\left[\begin{array}{cc} e^{-i\phi e} & 0\\ 0 & e^{-i\phi o} \end{array}\right]=\left[\begin{array}{cc} e^{-\frac{2\pi i}{\lambda }n_{e}L} & 0\\ 0 & e^{-\frac{2\pi i}{\lambda }n_{o}L} \end{array}\right]$$
and the below matricial equation:(6)$$\left[\begin{array}{c} E{{}^{\prime }}_{x}\\ E{{}^{\prime }}_{y} \end{array}\right]=\left[J\right]\cdot \left[\begin{array}{c} E_{x}\\ E_{y} \end{array}\right]=\left[\begin{array}{cc} e^{-\frac{2\pi i}{\lambda }n_{e}L} & 0\\ 0 & e^{-\frac{2\pi i}{\lambda }n_{o}L} \end{array}\right]\cdot \left[\begin{array}{c} E_{x}\\ E_{y} \end{array}\right]=\left[\begin{array}{c} E_{x}e^{-\frac{2\pi i}{\lambda }n_{e}L}\\ E_{y}e^{-\frac{2\pi i}{\lambda }n_{o}L} \end{array}\right]$$

The situation just described is ideal in the sense that we have considered a non-lossy perfectly transparent material. In reality, besides change of phase velocity due to the dielectric polarizability, the propagation of the light through such media is generally accompanied by optical attenuation [15]. This means that light wave loses a part of its energy due to various damping mechanisms, such as electronic transition absorption in the ultraviolet band, scattering of visible light mainly caused by imperfections and irregularities within the polymer, and phonon vibrational absorption in the infrared. In such cases, it is more convenient to use a complex refractive index (CRI) depicted by $\tilde{n}$ and defined as:(7)$$\tilde{n}=n-i\kappa$$
where its real part n is the medium refractive index and its imaginary part κ is the extinction coefficient representing the light wave absorbency of the lossy medium. In the particular case of a dichroic material, the absorption is different for light linearly polarized along the fast and slow axes and two distinct extinction coefficients, $\kappa_{e}$ and $\kappa_{o}$, are to be considered for each principal direction. As a result, one has to consider a different CRI for each of these directions as:(8)$${\tilde{n}}_{e,o}=n_{e,o}-i\kappa_{e,o}$$

Using the complex notation, the Jones matrix of the lossy uniaxial waveplate becomes [31]:$$\left[J\right]=\left[\begin{array}{cc} e^{-\frac{2\pi i(n_{e}-i\kappa_{e})}{\lambda }L} & 0\\ 0 & e^{-\frac{2\pi i(n_{o}-i\kappa_{o})}{\lambda }L} \end{array}\right]=\left[\begin{array}{cc} e^{-2\pi \kappa_{e}\frac{L}{\lambda }}e^{-i2\pi n_{e}\frac{L}{\lambda }} & 0\\ 0 & e^{-2\pi \kappa_{o}\frac{L}{\lambda }}e^{-i2\pi n_{o}\frac{L}{\lambda }} \end{array}\right]$$

Or simply,
(9)$$\left[J\right]=\left[\begin{array}{cc} t_{e}e^{-i\phi e} & 0\\ 0 & t_{o}e^{-i\phi o} \end{array}\right]$$
where $t_{e}=e^{-2\pi \kappa_{e}\frac{L}{\lambda }}$ and $t_{o}=e^{-2\pi \kappa_{o}\frac{L}{\lambda }}$ represent the principal amplitude transmittances of the uniaxial plane-parallel plate, respectively, in the extraordinary (*x*-axis) and ordinary (*y*-axis) directions. It is important to note that both $t_{e}$ and $t_{o}$ decay exponentially with the distance propagated, which corresponds to an attenuated wave traveling along the *z* direction. Another quantity, commonly used in spectrophotometry, to describe light absorbency of semi-transparent materials is the absorption coefficient α relating the transmitted light intensity $I{}^{\prime }$ after crossing a thickness L and the incident intensity I through the Beer–Lambert law [32]:(10)$$I{}^{\prime }=Ie^{-\alpha L}$$

Considering that light intensity is defined by the square of the magnitude of the Jones vector, transmitted intensities for both principal directions are calculated as:$$I_{e}^{\prime }={\left|\left[E_{e}^{\prime }\right]\right|}^{2}={\left|\left[J\right]\cdot \left[E_{e}\right]\right|}^{2}\;\mathrm{where}\;\left[E_{e}\right]=\left[\begin{array}{c} \sqrt{I}\\ 0 \end{array}\right]$$
$$I_{o}^{\prime }={\left|\left[E_{o}^{\prime }\right]\right|}^{2}={\left|\left[J\right]\cdot \left[E_{o}\right]\right|}^{2}\;\mathrm{where}\;\left[E_{o}\right]=\left[\begin{array}{c} 0\\ \sqrt{I} \end{array}\right]$$

Equation (10) can be rewritten for both directions by defining two different absorption coefficients $\alpha_{e}$ and $\alpha_{o}$ as:$$I_{e}^{\prime }=Ie^{-\alpha_{e}L}={\left|\left[J\right]\left[E_{e}\right]\right|}^{2}={\left|\left[\begin{array}{cc} e^{-2\pi \kappa_{e}\frac{L}{\lambda }}e^{-i2\pi n_{e}\frac{L}{\lambda }} & 0\\ 0 & e^{-2\pi \kappa_{o}\frac{L}{\lambda }}e^{-i2\pi n_{o}\frac{L}{\lambda }} \end{array}\right].\left[\begin{array}{c} \sqrt{I}\\ 0 \end{array}\right]\right|}^{2}=Ie^{-4\pi \kappa_{e}\frac{L}{\lambda }}$$
$$I_{o}^{\prime }=Ie^{-\alpha_{o}L}={\left|\left[J\right]\left[E_{o}\right]\right|}^{2}={\left|\left[\begin{array}{cc} e^{-2\pi \kappa_{e}\frac{L}{\lambda }}e^{-i2\pi n_{e}\frac{L}{\lambda }} & 0\\ 0 & e^{-2\pi \kappa_{o}\frac{L}{\lambda }}e^{-i2\pi n_{o}\frac{L}{\lambda }} \end{array}\right].\left[\begin{array}{c} 0\\ \sqrt{I} \end{array}\right]\right|}^{2}=Ie^{-4\pi \kappa_{o}\frac{L}{\lambda }}$$

This enables one to derive the following relation between the absorption coefficients and the extinction coefficients:(11)$$\alpha_{e,o}=\frac{4\pi }{\lambda }\kappa_{e,o}$$

### 2.2. Transmission and Reflection Coefficients

The analytical modeling described in the previous section takes into account the attenuation due to absorption inside the material without considering all successive multiple reflections happening at the interfaces as predicted by the boundary conditions. The expressions of the amplitude transmittances $t_{e}$ and $t_{o}$ presented in Equation (9) need to be modified to include Fresnel’s coefficients. The intensity reflectance and transmittance of a plane wave at the interface between air and a partially absorbing dielectric plane-parallel plate at normal incidence can be found in any standard optics textbook and evaluated by replacing the refractive index in the expression of the Fresnel’s coefficients with its complex counterpart [33]:(12)$$R_{int}={\left|\frac{\tilde{n}-1}{\tilde{n}+1}\right|}^{2}={\left|\frac{n-i\kappa -1}{n-i\kappa +1}\right|}^{2}=\frac{{\left(n-1\right)}^{2}+\kappa^{2}}{{\left(n+1\right)}^{2}+\kappa^{2}}\;\mathrm{Interface}\;\mathrm{intensity}\;\mathrm{reflectance}$$
(13)$$T_{int}=1-R_{int}=1-\frac{{\left(n-1\right)}^{2}+\kappa^{2}}{{\left(n+1\right)}^{2}+\kappa^{2}}=\frac{4n}{{\left(n+1\right)}^{2}+\kappa^{2}}\;\mathrm{Interface}\;\mathrm{intensity}\;\mathrm{transmittance}$$

Note that in the above equations we assumed no absorption at the interfaces (as opposed to absorption in the bulk due to propagation) and adopted the hypothesis of an ideally flat plate surface to eliminate any diffuse reflectance or transmittance from the interfaces, which implies that $R_{int}$ and $T_{int}$ should necessarily sum up to 1. Considering the plate faces to be two perfectly identical semi-reflective surfaces, the entire intensity reflection and transmission coefficients R and T can be calculated from Equations (12) and (13) with infinite summations over the multi-reflected contributions of the faces. Moreover, in the case of an absorbing medium, any internal reflection is necessarily preceded by a penetration through the plate, and, therefore, a light intensity damping factor per penetration equal to ($e^{-\alpha d}$) has to be considered in this calculation (see Figure 1):$$R=R_{int}+R_{int}T_{int}^{2}e^{-2\alpha d}+R_{int}^{3}T_{int}^{2}e^{-4\alpha d}+R_{int}^{5}T_{int}^{2}e^{-6\alpha d}+\ldots$$
$$T=T_{int}^{2}e^{-\alpha d}+T_{int}^{2}R_{int}^{2}e^{-3\alpha d}+T_{int}^{2}R_{int}^{4}e^{-5\alpha d}+T_{int}^{2}R_{int}^{6}e^{-7\alpha d}+\ldots$$
$$A=T_{int}\left(1-e^{-\alpha d}\right)+R_{int}T_{int}e^{-\alpha d}\left(1-e^{-\alpha d}\right)+R_{int}^{2}T_{int}e^{-2\alpha d}\left(1-e^{-\alpha d}\right)+\ldots$$

It is immediately seen from the previous formulae that *R*, *T*, and *A* expressions represent the infinite sums of geometric series with a common ratio $q={\left[R_{int}e^{-\alpha d}\right]}^{2}$ for R,T and $q^{\prime }=\left[R_{int}e^{-\alpha d}\right]$ for *A*:$$R=R_{int}+R_{int}T_{int}^{2}e^{-2\alpha d}\overset{\infty }{\underset{k=0}{\sum }}{\left[R_{int}e^{-\alpha d}\right]}^{2k}$$
$$T=T_{int}^{2}e^{-\alpha d}\overset{\infty }{\underset{k=0}{\sum }}{\left[R_{int}e^{-\alpha d}\right]}^{2k}$$
$$A=T_{int}\left(1-e^{-\alpha d}\right)\overset{\infty }{\underset{k=0}{\sum }}{\left[R_{int}e^{-\alpha d}\right]}^{k}$$

With the convergence condition $\left|q\right|<1$ being met, these sums lead to the formulae:(14)$$R=R_{int}+\frac{R_{int}T_{int}^{2}e^{-2\alpha d}}{1-{\left(R_{int}e^{-\alpha d}\right)}^{2}}$$
(15)$$T=\frac{T_{int}^{2}e^{-\alpha d}}{1-{\left(R_{int}e^{-\alpha d}\right)}^{2}}$$
(16)$$A=\frac{T_{int}\left(1-e^{-\alpha d}\right)}{1-R_{int}e^{-\alpha d}}$$

By adding the above three expressions, using some mathematical manipulations, and recalling that $R_{int}+T_{int}=1$, it can be verified that the energy conservation requirement is fulfilled:R+T+A=1

It is important to mention that the above summations could not be achieved without the use of the slab approximation for which the plane-parallel plate is assumed to be optically thick. In other words, the optical path length nd through the plate is much larger than the incident light coherence length $L_{c}$ defined by the formula [34,35]:(17)$$L_{c}=\frac{\lambda^{2}}{2\pi \Delta \lambda }$$
where Δλ represents the wavelength resolution (bandwidth) of the spectrophotometer radiation source. In the case where $nd\gg L_{c}$, the phases of the internal multiple reflected light waves are randomized, leading to incoherent summations of waves that will be averaged out and, hence, cannot interfere with each other [33,34,35].

Using the previous rigorous formulation taking into account both interfaces and bulk contributions, the principal intensity transmittances of the absorbing uniaxial slab could be derived by substituting Equations (12) and (13) into Equation (15) as follows:(18)$${\left|t_{o,e}\right|}^{2}=T_{o,e}=\frac{{\left(\frac{4n_{o,e}}{{\left(n_{o,e}+1\right)}^{2}+\kappa_{o,e}^{2}}\right)}^{2}e^{-\left(\frac{4\pi d}{\lambda }\right)\kappa_{o,e}}}{1-{\left(\frac{{\left(n_{o,e}-1\right)}^{2}+\kappa_{o,e}^{2}}{{\left(n_{o,e}+1\right)}^{2}+\kappa_{o,e}^{2}}\;\right)}^{2}e^{-2\left(\frac{4\pi d}{\lambda }\right)\kappa_{o,e}}}$$

### 2.3. Cauchy Model of Dispersion

The visible band dispersion curve observation of various typical transparent materials used in optics and photonics shows a common characteristic of normal dispersion for which the index of refraction decreases as the wavelength increases with a lower rate of decrease at higher wavelengths [36], which could be mathematically expressed as follows:(19)$$\frac{dn}{d\lambda }<0\;\mathrm{and}\;\frac{d^{2}n}{d\lambda^{2}}>0$$

Several empirical models have been suggested to formulate the normal dispersion property and describe the wavelength-dependent response of dielectric media for light propagation. Among these, the Cauchy formula is widely in use due to its simplicity, intuitiveness, and success in accurately modeling most of the known transparent material visible dispersion. It is commonly stated in the following form [15,36,37]:(20)$$n\left(\lambda \right)=A+\frac{B}{\lambda^{2}}+\frac{C}{\lambda^{4}}$$
where A, B, and C are constants characteristic of the material. In this model, the constant A defines the index amplitude, while B and C add curvature to produce normal dispersion. With wavelength units in nanometers, the typical range for B and C is [18]:(21)$$10^{3}<B\left(nm^{2}\right)<5.10^{4}\;\mathrm{and}-10^{9}<C\left(nm^{4}\right)<5.10^{9}$$

The Cauchy dispersion equation was originally used for material with no optical absorption, and, in general, it works best far from any absorption band. However, in the case of weakly absorbing dielectrics, the formula can be extended to cover the absorbing regime [37], using an additional formula for $\kappa \left(\lambda \right)$:(22a)$$n\left(\lambda \right)=A_{n}+\frac{B_{n}}{\lambda^{2}}+\frac{C_{n}}{\lambda^{4}}$$
(22b)$$\kappa \left(\lambda \right)=A_{\kappa }+\frac{B_{\kappa }}{\lambda^{2}}+\frac{C_{\kappa }}{\lambda^{4}}\;$$

Recently reported spectroscopic ellipsometry measurement of PET films in the visible range [11] has revealed a normal dispersion behavior of the extinction coefficient which justifies the use of the “Cauchy absorbing” model in the present work. Note that the coefficients of Equation (22b) are analogous and behave similarly to those in (22a), with a typical range of [37]:(23)$$-10^{4}<B_{\kappa }\left(nm^{2}\right)<10^{4}\;\mathrm{and}-10^{9}<C_{\kappa }\left(nm^{4}\right)<10^{9}$$

## 3. Materials and Methods

### 3.1. The Sample

The studied sample is a 1 × 2 cm^2^ rectangular sheet cut from a commercially available Mylar film (Mylar^®^ X6739, Gauge 750, Teijin-Dupont, Chester, VA, USA) such that one side of the rectangle was parallel to the optical axis direction. A higher accurate value of sheet thickness estimated to d = (166 ± 1) μm was obtained using a white-light Michelson interferometric method described elsewhere [38,39]. Mylar is the trade name of a well-known family of polyester films made from biaxially oriented polyethylene terephthalate (BOPET). Its chemical structure is described in (Figure 2) with a phenyl ring and an ester group.

At the beginning of the manufacturing process, the PET film has a semicrystalline structure composed of both crystalline and amorphous phases [1,4,41]. The size of these crystallites, commonly called spherulites, is generally of the same order of magnitude as the wavelength of visible light, which causes it to scatter at the boundaries between these crystallites and the amorphous regions, resulting in its poor transparency [41]. The PET film is then biaxially stretched along the machine direction (MD) and the transverse direction (TD) through a simultaneous or sequential drawing process with a typical draw ratio of about 3 to 4 in both directions [3,42,43]. The drawing has two main effects on the structure and morphology of the PET film; on the one hand, the randomly distributed PET polymeric chains tend to preferentially align along the stretch direction [43], which induces anisotropy in its mechanical and optical properties. On the other hand, stretching reduces the dimensions of the spherulites [41], which significantly improves the PET film transparency and stiffness but enhances its commercial value.

### 3.2. Transmission Spectroscopy Setup

The Mylar film is placed between two identical dichroic sheet polarizers (P) and (A) in two different configurations (see Figure 3): (1) polarizer and analyzer transmission axes and the Mylar optical axis are all oriented along the x-direction (extraordinary transmittance configuration); (2) polarizer and analyzer orientation is kept the same, while the Mylar sample optical axis is aligned along the y-direction (ordinary transmittance configuration). To ensure transmittance and thickness uniformity, (P) and (A) were cut from the same original Polaroid sheet. We used the XP38 Polaroid from Edmund Optics [44] with an extinction ratio up to 7500:1 in the visible range. The intensity transmittance spectra were recorded using a Perkin-Elmer Lambda 950 double beam and dual monochromator UV/VIS/NIR spectrophotometer in the wavelength interval from 400 nm to 750 nm with steps of 5 nm.

As depicted schematically in (Figure 4), a white light radiation emitted from a halogen lamp (HL) is directed by reflective mirrors through the entrance slit of the primary monochromator (MCH1) then dispersed by a first holographic grating (G1) with 1440 lines/mm blazed at 240 nm. During the scanning operation, the monochromator slewing mechanism selects a specific wavelength segment from the dispersed light and reflects it through the exit slit of (MCH1) in the form of a near-monochromatic radiation beam. In order to achieve high spectral purity with an extremely low stray light, the beam exiting (MCH1) is redirected to the entrance slit of a secondary monochromator (MCH2) for further spectral refinement. The output beam from (MCH2) is then split by a chopper (CHP) with a 46 Hz cycle rotating mechanism into a reference beam (RB) and a sample beam (SB). After crossing the sample compartment, the light intensities of both beams are alternately measured by a high sensibility R6872 photomultiplier detector (PHM). It should be noted that white light emitted from the halogen lamp was initially unpolarized by nature. However, it becomes partially polarized because of the multiple interactions that it undergoes with the spectrophotometer’s optical components (diffraction by the monochromators’ slits and gratings, reflections by mirrors). To overcome this shortcoming, a depolarizer drive accessory [45] is mounted at the entrance of the sample compartment, thereby ensuring a depolarization efficiency greater than 98% for both reference and sample beams.

### 3.3. Polarizer Transmittance

The first step of the experimental procedure is to determine the transmittance of the used polarizers over the working spectral range. To achieve this, both (P) and (A) are placed in the Lambda 950 sample compartment without the Mylar sheet, and their intensity transmittance spectra $T_{∥}\left(\lambda \right)$ and $T_{⊥}\left(\lambda \right)$ corresponding, respectively, to the parallel and crossed configurations are recorded (see Figure 5). Dichroic polymer polarizers are known to be non-ideal. Consequently, they are characterized by two principal intensity transmittances $T_{x}$ and $T_{y}$. Assuming *x*-axis to be the polarizer transmitting direction, we have [47]:
(24)$$T_{x}=\frac{1}{\sqrt{2}}\left[{\left(T_{∥}+T_{⊥}\right)}^{\frac{1}{2}}+{\left(T_{∥}-T_{⊥}\right)}^{\frac{1}{2}}\;\right]$$
(25)$$T_{y}=\frac{1}{\sqrt{2}}\left[{\left(T_{∥}+T_{⊥}\right)}^{\frac{1}{2}}-{\left(T_{∥}-T_{⊥}\right)}^{\frac{1}{2}}\;\right]$$

It can be seen from Figure 5b that instead of a zero transmittance as in an ideal pair of crossed polarizers, one still detects some amount of transmitted light that could be neglected ($T_{⊥}\ll T_{∥}$). This approximation enables simplifying Equations (24) and (25) without appreciable loss of accuracy, leading to:(26)$$T_{x}={\left(2T_{∥}\right)}^{\frac{1}{2}}$$
(27)$$T_{y}=0$$

### 3.4. Modeling and Fitting

Using the partially absorbing uniaxial waveplate model described in Section 2.1, the system Polarizer-Mylar-Analyzer (PMA) for each configuration can be described by the below Jones matrices:$$\left[J_{PMA}^{o}\right]=\left[J_{A}\right].\left[J_{Mylar}^{o}\right].\left[J_{P}\right]\;\mathrm{Ordinary}\;\mathrm{transmittance}\;\mathrm{configuration}\;$$
$$\left[J_{PMA}^{e}\right]=\left[J_{A}\right].\left[J_{Mylar}^{e}\right].\left[J_{P}\right]\;\mathrm{Extraordinary}\;\mathrm{transmittance}\;\mathrm{configuration}$$
$$\mathrm{where}\;\left\{\begin{array}{c} \left[J_{Mylar}^{o}\right]=\left[\begin{array}{cc} t_{o}e^{i\varphi_{o}}\; & 0\\ 0 & t_{e}e^{i\varphi_{e}} \end{array}\right]\\ \left[J_{Mylar}^{e}\right]=\left[\begin{array}{cc} t_{e}e^{i\varphi_{e}}\; & 0\\ 0 & t_{o}e^{i\varphi_{o}} \end{array}\right]\\ \left[J_{P}\right]=\left[J_{A}\right]=\left[\begin{array}{cc} \sqrt{T_{x}} & 0\\ 0 & 0 \end{array}\right] \end{array}\right.$$

Combining the above expressions leads to:(28a)$$\left[J_{PMA}^{o}\right]=\left[\begin{array}{cc} T_{x}t_{o}e^{i\varphi_{o}}\; & 0\\ 0 & 0 \end{array}\right]$$
(28b)$$\left[J_{PMA}^{e}\right]=\left[\begin{array}{cc} T_{x}t_{e}e^{i\varphi_{e}}\; & 0\\ 0 & 0 \end{array}\right]$$

According to the Jones calculus [48], intensity transmittance of unpolarized light through an optical system described with its Jones matrix $\left[J\right]$ is given by:(29)$$J=\left[\begin{array}{cc} J_{11} & J_{21}\\ J_{12} & J_{22} \end{array}\right]\Rightarrow T_{unpolarized}=\frac{1}{2}\underset{ij}{\sum }{\left|J_{ij}\right|}^{2}=\frac{1}{2}\left({\left|J_{11}\right|}^{2}+{\left|J_{21}\right|}^{2}+{\left|J_{12}\right|}^{2}+{\left|J_{22}\right|}^{2}\right)$$

Using Equation (29), one can easily derive the intensity transmittance of the PMA system in both configurations and obtains:(30a)$$T_{PMA}^{o}=\frac{1}{2}{\left|T_{x}t_{o}e^{i\varphi_{o}}\right|}^{2}=\frac{T_{x}^{2}{\left|t_{o}\right|}^{2}}{2}=\frac{T_{x}^{2}}{2}T_{o}$$
(30b)$$T_{PMA}^{e}=\frac{1}{2}{\left|T_{x}t_{e}e^{i\varphi_{e}}\right|}^{2}=\frac{T_{x}^{2}{\left|t_{e}\right|}^{2}}{2}=\frac{T_{x}^{2}}{2}T_{e}\;\;$$

Mylar sample ordinary and extraordinary intensity transmittance dispersion curves could be finally extracted from the recorded spectra as follows:(31a)$$T_{o}^{exp}\left(\lambda \right)=\frac{2T_{PMA}^{o}\left(\lambda \right)}{{\left[T_{x}\left(\lambda \right)\right]}^{2}}$$
(31b)$$T_{e}^{exp}\left(\lambda \right)=\frac{2T_{PMA}^{e}\left(\lambda \right)}{{\left[T_{x}\left(\lambda \right)\right]}^{2}}\;$$

The polarizer principal transmittance dispersion curve $T_{x}\left(\lambda \right)$ shown in Figure 6 was calculated using Equation (26). The two dispersion spectra $T_{PMA}^{o}\left(\lambda \right)$ and $T_{PMA}^{e}\left(\lambda \right)$ of the system Polarizer-Mylar-Analyzer were recorded using the experimental setup described in Section 3.2 and shown in Figure 7.

Finally, the ordinary and extraordinary transmittances were calculated using Equation (31a,b) and plotted in the same graph (Figure 8). It should be noted that the used spectrophotometer secures a spectral resolution of Δλ=5 nm over the visible range, resulting in a coherence length ranging between 5 μm and 18 μm as defined in the Formula (17). Therefore, the thick slab approximation can be reasonably used to model the 166 μm thick Mylar sample as explained previously.

The spectrophotometric technique presented in the current work is based on a curve-fitting approach conducted over the entire wavelength range between the measured sample transmittances $T_{o,e}^{exp}\left(\lambda \right)$ and the analytical model defined in Equation (18). In this formula, the transmittance wavelength dependence is achieved through the use of the Cauchy dispersion formula for both $n\left(\lambda \right)$ and $\kappa \left(\lambda \right)$. Now, by substituting Equation (22) into Equation (18), one obtains:(32)$$T_{o,e}\left(\lambda \right)=\frac{{\left[\frac{4\left(A_{n}^{o,e}+\frac{B_{n}^{o,e}}{\lambda^{2}}+\frac{C_{n}^{o,e}}{\lambda^{4}}\right)}{{\left(A_{n}^{o,e}+1+\frac{B_{n}^{o,e}}{\lambda^{2}}+\frac{C_{n}^{o,e}}{\lambda^{4}}\right)}^{2}+{\left(A_{\kappa }^{o,e}+\frac{B_{\kappa }^{o,e}}{\lambda^{2}}+\frac{C_{\kappa }^{o,e}}{\lambda^{4}}\right)}^{2}}\right]}^{2}e^{-\left(\frac{4\pi d}{\lambda }\right)\left(A_{\kappa }^{o,e}+\frac{B_{\kappa }^{o,e}}{\lambda^{2}}+\frac{C_{\kappa }^{o,e}}{\lambda^{4}}\right)}}{1-{\left[\frac{{\left(A_{n}^{o,e}-1+\frac{B_{n}^{o,e}}{\lambda^{2}}+\frac{C_{n}^{o,e}}{\lambda^{4}}\right)}^{2}+{\left(A_{\kappa }^{o,e}+\frac{B_{\kappa }^{o,e}}{\lambda^{2}}+\frac{C_{\kappa }^{o,e}}{\lambda^{4}}\right)}^{2}}{{\left(A_{n}^{o,e}+1+\frac{B_{n}^{o,e}}{\lambda^{2}}+\frac{C_{n}^{o,e}}{\lambda^{4}}\right)}^{2}+{\left(A_{\kappa }^{o,e}+\frac{B_{\kappa }^{o,e}}{\lambda^{2}}+\frac{C_{\kappa }^{o,e}}{\lambda^{4}}\right)}^{2}}\right]}^{2}\;e^{-2\left(\frac{4\pi d}{\lambda }\right)\left(A_{\kappa }^{o,e}+\frac{B_{\kappa }^{o,e}}{\lambda^{2}}+\frac{C_{\kappa }^{o,e}}{\lambda^{4}}\right)}}$$

All data fits were performed using the commercial package Mathematica [49] through its NonlinearModelFit [50] built-in function. The non-linear fitting aims to estimate the values of the Cauchy equation parameters $A_{n,\kappa }^{o,e}$, $B_{n,\kappa }^{o,e}$, and $C_{n,\kappa }^{o,e}$ defined in the Formula (32) which best describe the experimental intensity transmittance dispersion curves $T_{o,e}^{exp}\left(\lambda \right)$. NonlinearModelFit uses a standard Non-Linear Least-Squares Fitting (NLSF) algorithm consisting of iteratively computing and minimizing the deviations (variance) of the theoretical curves from the experimental data points described by the quantity chi-squared ($\chi^{2}$). Note that in the present work, minimizations were conducted under the assumption that experimental data were independently and normally distributed with the same standard deviation, leading to all experimental data points with the same statistical weight, set to 1, which gives a more simplified formula for $\chi^{2}$, commonly called residual sum of squares (RSS) and defined as:(33)$$\chi^{2}=\overset{750}{\underset{\lambda_{i}=400}{\sum }}{\left[T_{o,e}^{exp}\left(\lambda_{i}\right)-T_{o,e}\left(\lambda_{i}\right)\right]}^{2}$$

Additional constraints were added to the fitting function in order to obtain physically acceptable solutions, guaranteeing normal dispersion for both $n_{o,e}\left(\lambda \right)$ and $\kappa_{o,e}\left(\lambda \right)$, as defined in Equation (19), along with a positiveness for the quantities $\Delta n\left(\lambda \right)$ and $\Delta \kappa \left(\lambda \right)$ over the whole working spectral range. During each iteration, NonlinearModelFit uses a standard Wolfram constrained optimization procedure [51] that automatically selects the best method to be used to adjust the values of the 12 fitting parameters. Depending on the closeness of the initial estimates to the optimal desired solution, the Wolfram optimization procedure switches between the Levenberg–Marquardt [52] and the Quasi-Newton [53] methods to ensure a fast, reliable, and optimal convergence [54]. This iterative process completes (converges) when the difference between two consecutive calculated $\chi^{2}$ is less than a preset tolerance value defined as the convergence criterion. It is worth mentioning that (NLSF) optimizations are highly sensitive to the choice of the initial guess values. In other words, convergence to optimal solutions is only obtained for a narrow range of guess values. This disadvantage was overcome by repeating the fit several times while varying the initial parameters through a marching procedure to find the best fit. In order to ensure a fitting convergence to an acceptable solution with a reduced number of iterations, the starting guess parameters were estimated using a three-step procedure described in detail in Appendix A.

In addition to a minimal value of $\chi^{2}$, the goodness of fit was evaluated by two other indicators: the root mean-square error (RMSE) and $R^{2}$ value, also known as the coefficient of determination (COD). Let us recall that a better fit is indicated by a high value (close to 1) of $R^{2}$ and a low value (close to 0) of both $\chi^{2}$ and (RMSE).

## 4. Results and Discussion

Table 1 presents the obtained fitting results of both $T_{o}^{exp}\left(\lambda \right)$ and $T_{e}^{exp}\left(\lambda \right)$ spectra and their corresponding goodness-of-fit indicators (GFI). It is immediately visible that the values of the 12 parameters $A_{n,\kappa }^{o,e}$, $B_{n,\kappa }^{o,e}$, and $C_{n,\kappa }^{o,e}$ fall well within the experimental ranges previously defined in Equations (21) and (23). The theoretical intensity transmittances of the partially absorbing uniaxial slab model in Equation (18) display a good agreement with the experimental curves for both ordinary and extraordinary configurations as shown in Figure 9, and this is attested by the obtained values of $\chi^{2}$, $R^{2}$, and RMSE (see Table 1).

It is important to remember that Cauchy’s coefficients $A_{n,\kappa }^{o,e}$, $B_{n,\kappa }^{o,e}$, and $C_{n,\kappa }^{o,e}$ are independently adjustable parameters used to empirically describe the sample dispersion function and, therefore, have no physical meaning. While coefficient $A_{n,\kappa }^{o,e}$ defines the range of $n_{e,o}\left(\lambda \right)$ and $\kappa_{e,o}\left(\lambda \right)$ and should be necessarily positive, $B_{n,\kappa }^{o,e}$, and $C_{n,\kappa }^{o,e}$ adjust the curvature and, hence, may be negative numbers. Figure 9 illustrates a typical increase of transmittances with the wavelength. The Mylar sample has maximum transparency in the direction of the NIR region of the spectrum, with a maximum transmittance of about 0.9. It gradually tends to become translucent for the smaller wavelengths, with a minimum transmittance around 0.4. This dispersion behavior is observed for both ordinary and extraordinary polarized incident lights, with a negative differential transmittance also known as diattenuation ($\Delta T=T_{e}-T_{o}<0$).

The diattenuation $\left|\Delta T\right|$ order of magnitude is remarkably significant, roughly ranging between 0.02 and 0.04 and corresponding to an average ratio of about 4.6%. This amount of discrepancy could not solely be explained by the birefringence (Δn) of the Mylar sample but also requires considering non-negligible extinction coefficients. An approximate estimation of $\left|\Delta T\right|$ can be obtained (see Appendix B), assuming a non-lossy sample leading to the following upper bound:(34)$$\left|\Delta T\right|\leq 0.25\times \Delta n$$

Recent spectroscopic works on Mylar films [1,2,55] reported a birefringence value between 0.04 and 0.06 in the (400, 750) nm spectral band. Using the condition in Equation (34) and the experimental range of Δn, the expected value of $\left|\Delta T\right|$ is estimated to be ranging between 0.01 and 0.015, which is nearly half of the observed value. It was this discrepancy between observed and expected values of the diattenuation that led us, in the first place, to suggest the partially absorbing uniaxial slab model with a complex refractive index in this research study. In a second step, the refractive indices dispersion curves $n_{o}\left(\lambda \right)$ and $n_{e}\left(\lambda \right)$ along with the birefringence $\Delta n\left(\lambda \right)$ are calculated and plotted using the best fit parameters $A_{n}^{o,e}$, $B_{n}^{o,e}$, and $C_{n}^{o,e}$ presented in Table 1. As can be seen from Figure 10, the calculated indices show normal dispersion with values ranging between 1.55 and 1.75. On the other hand, the comparison depicted in Figure 11 shows that values of $\Delta n\left(\lambda \right)$ obtained by our method are slightly larger than those reported in the literature [1,2]. It is conceivable that analyzed PET samples in these different experimental works received different drawing ratios during their biaxial stretching process and, hence, acquired different amounts of birefringence as proven by Cakmak et al. [56].

The extinction coefficients dispersion curves are, finally, deduced from the six remaining best fit parameters of Table 1. The wavelength dependence of $\kappa_{o}$ and $\kappa_{e}$ presented in Figure 12 shows a normal dispersion curvature for both ordinary and extraordinary polarizations, with values decreasing from $15\times 10^{-5}$ to $1.0\times 10^{-5}$.

In addition, a very small positive extinction coefficient difference $\Delta \kappa =\kappa_{e}-\kappa_{o}$ analogous to the birefringence is observed. In turn, the shift Δκ exhibits normal dispersion and ranges between $5.0\times 10^{-6}$ and $15\times 10^{-6}$ as shown in Figure 13. The reported discrepancy Δκ is a proof that stretched PET films display a small amount of dichroism that contributes with the birefringence Δn to the observed diattenuation $\left|\Delta T\right|$ between ordinary and extraordinary transmittances.

These findings enable further simplifications of Formula (18) employing reasonable approximations (see Appendix C) as follows:(35)$$T\left(\lambda \right)=\frac{16n^{2}}{{\left(n+1\right)}^{4}}e^{-\left(\frac{4\pi d}{\lambda }\right)\kappa }$$

The last equation shows that Mylar film transmittance for a given wavelength is the product of two terms corresponding to the contribution of real and imaginary parts of the complex refractive index and defined, respectively, as:(36)$$T_{n}\left(\lambda \right)=\frac{16n^{2}}{{\left(n+1\right)}^{4}}\;T_{\kappa }\left(\lambda \right)=e^{-\left(\frac{4\pi d}{\lambda }\right)\kappa }$$

The extinction coefficients values of the current study were compared to the results obtained by spectroscopic ellipsometry in reference [11] and plotted together on a logarithmic scale (see Figure 14). It can be immediately seen that values of $\kappa \left(\lambda \right)$ of this work are approximately two orders of magnitude larger than those reported in [11]. The difference could be explained by the use of different types of Mylar sheets in the two experiments. Commercially available Mylar polyester sheets present different degrees of transmission haze, defined as the percentage of diffusely transmitted light scattered at larger angles (>2.5°) [41].

Depending on their type, some films are completely transparent and transmit visible light very similar to window glass, such as Mylar type-D sheets [57] with thickness of 70 µm to 250 µm (0% < haze < 2%), while others are extremely hazy, such as Mylar type-A sheets [58] with thickness of 250 µm and above (90% < haze < 100%). It should be noted that the difference in haze level is attributed to two factors that increase the photon scattering at interfaces [59]: the surface roughness of the film and the discontinuities of refractive index between spherulites due to the semicrystalline structure of PET films. As mentioned previously, the Mylar^®^ X6739 sheet (Chester, VA, USA) used in the current work has a thickness of 166 µm. Therefore, according to reference [60], its haze value should be around 20%, which makes it slightly translucent and gives it a milky appearance in visible light. According to the reported $\kappa \left(\lambda \right)$ order of magnitude in reference [11], it can be assumed that a PET sample with higher transparency was used in that experiment, such as Mylar type-D or type-C (haze about 5%). Based on this assumption, the visible light transmittance ratio of the two experiments is expected to be around 95:80 ≈1.2.

To confirm this interpretation, the PET sample transmittance $T_{11}\left(\lambda \right)$ of reference [11] was calculated for each value of the wavelength using the Formula (35) and the dispersion functions $n\left(\lambda \right)$ and $\kappa \left(\lambda \right)$ reported in the same reference. $T_{11}\left(\lambda \right)$ values are then compared with the fitted transmittances $T_{o}\left(\lambda \right)$ and $T_{e}\left(\lambda \right)$ of the current study. As can be seen in Figure 15, the dispersion curve $T_{11}\left(\lambda \right)$ is nearly constant across the working visible range, with an average value around 0.9. Very similar transmittance curves were reported for Mylar type-C and type-D in Dupont Teijin datasheets [61], confirming our assumption. Moreover, the average ratios $T_{11}\left(\lambda \right):\;T_{o}\left(\lambda \right)$ and $T_{11}\left(\lambda \right):\;T_{e}\left(\lambda \right)$ were calculated and found close to the predicted value of 95:80 and, respectively, equal to 1.23 and 1.29.

These findings suggest that visible light extinction through a stretched PET film is primarily due to the effect of haze. Our calculations revealed that Mylar sheet’s extinction coefficient is dominantly controlled by the amount of diffusively transmitted light and that the visible light absorption effect is negligible compared to scattering. To our knowledge, no experimental or theoretical works have reported, so far, any correlation between incident light polarization and haze value. This leads us to admit that Δκ is only attributable to the difference of absorption coefficient between the two principal polarizations. The influence of dichroism in stretched PET films cannot simply be ignored, even if its impact on the extinction coefficient value has been deemed insignificant.

Finally, it should be noted that the proposed method gives reliable results only in the absence of coherent superposition of multiple internally reflected and transmitted waves. This condition is guaranteed by using an optically thick sample for which the coherence length $L_{c}$ is small enough compared to the thickness d. It is essential to define a lower limit $d_{min}$ of sample thickness below which the transmission Formula (18) has to be modified to include the interference terms. The minimum allowable thickness $d_{min}$ is mainly determined by a couple of factors: first, the spectral bandwidth Δλ of the used spectrophotometer, depending essentially on the physical width of the monochromator slits; second, the wavelength working range and, more precisely, its highest value $\lambda_{max}$. A thickness criterion can be established based on definition (17) of the coherence length that has to be at least one order of magnitude smaller than d as follows:(37)$$d_{min}=10\times \frac{\lambda_{max\;}}{2\pi \Delta \lambda }\;\mathrm{Or}\;\mathrm{simply}\;d_{min}\;\sim \frac{\lambda_{max\;}}{\Delta \lambda }$$

Condition (37) shows that for larger wavelengths, in the IR region of the spectrum, the critical thickness $d_{min}$ increases in a way that limits the range of allowable thicknesses. In that case, the technique has to be modified in order to take into consideration both coherent and incoherent superpositions.

## 5. Conclusions

In this work, we have developed a new technique to extract the dispersion curves of a Mylar film’s ordinary and extraordinary complex refractive indices (CRIs) in the visible band. The suggested method is fast, straightforward, and easy to set up and presents the advantage of using low-cost instrumentation (double-beam spectrophotometer). It enables the determination of both the refractive index and extinction coefficient of a uniaxial sample with known thickness by fitting a recorded transmittance spectrum obtained by spectrophotometry with a theoretical model that we had elaborated in advance. We used a set of two spectra corresponding to ordinary and extraordinary transmittances, allowing a simultaneous determination of the ordinary and extraordinary CRIs. We derived a new generalized formula for transmittance dispersion based on the Fresnel’s coefficients formalism and using the Cauchy model to describe n and κ wavelength dependence. Our model was able to reproduce the Mylar transmission spectra in both configurations, and the retrieved dispersion curves $n_{o,e}\left(\lambda \right)$, $\kappa_{o,e}\left(\lambda \right)$, and $\Delta n\left(\lambda \right)$ are comparable to corresponding ones found in the literature. Our experimental approach demonstrated its ability to detect and estimate the discrepancy $\Delta \kappa \left(\lambda \right)$ of highly transparent media such as Mylar films across the visible band, often inaccessible through other techniques such as spectroscopic ellipsometry. With the obtained values of $\Delta \kappa \left(\lambda \right)$, we have proved that, in addition to the birefringence, Mylar films exhibit a very small amount of dichroism. Using several approximations, we succeeded in further simplifying the transmission dispersion formula and confirmed that the Mylar sheet extinction coefficient is dominantly controlled by the amount of haze due to diffusively transmitted light. Finally, it is important to note that Mylar is found to exhibit a similar birefringence dispersion over a wide spectral range to liquid crystals, employed for display applications, which makes our method applicable for liquid crystal optical anisotropy investigation.

## Figures and Tables

**Figure 1 polymers-14-01805-f001:**
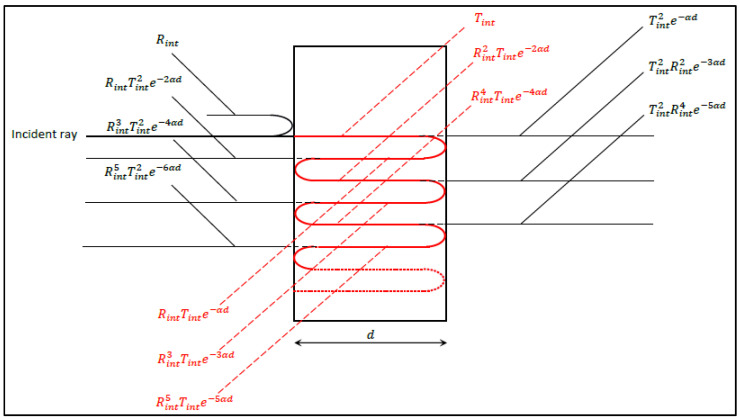
Multiple ray reflections and transmissions of a normally incident light beam inside an optically thick plane-parallel dielectric plate.

**Figure 2 polymers-14-01805-f002:**
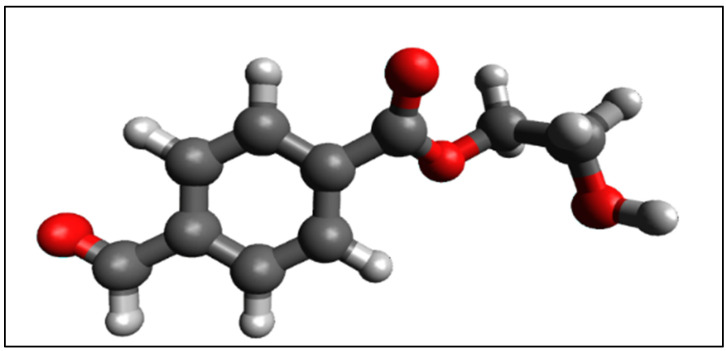
3D model of the PET molecular structure. The polymeric chain is formed by linking successively this monomer (repeat unit) about 100 times on average. The model was produced by the Avogadro molecule editor [40].

**Figure 3 polymers-14-01805-f003:**
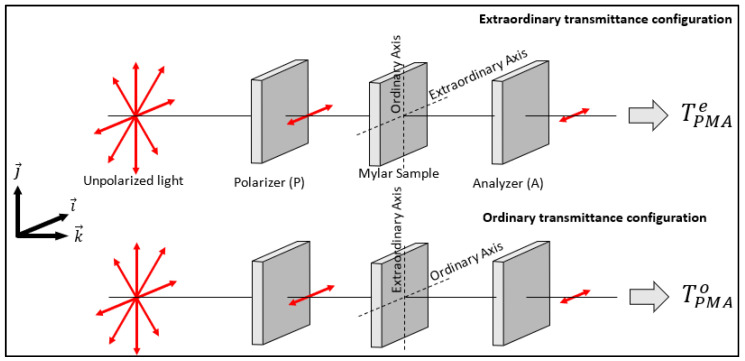
Experimental setup with the ordinary and extraordinary transmittance measurement configurations. The system Polarizer-Mylar-Analyzer (PMA) is placed in the sample compartment of the spectrophotometer.

**Figure 4 polymers-14-01805-f004:**
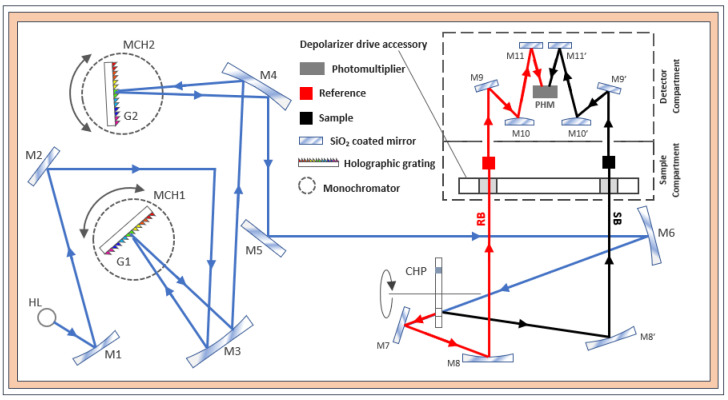
Schematic representation of Lambda 950 transmission spectrophotometer setup redrawn and adapted from reference [46].

**Figure 5 polymers-14-01805-f005:**
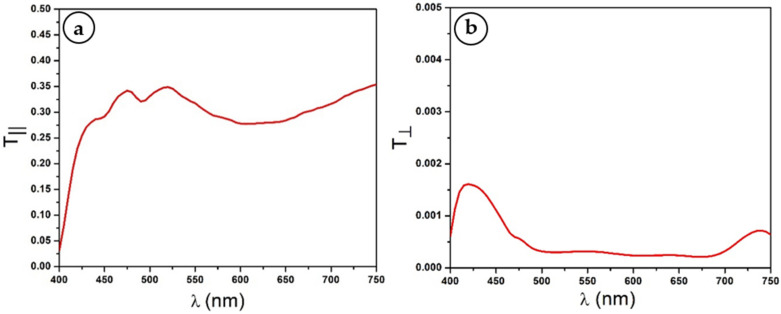
Polarizer and analyzer transmittance spectra for both parallel (**a**) and crossed (**b**) configurations.

**Figure 6 polymers-14-01805-f006:**
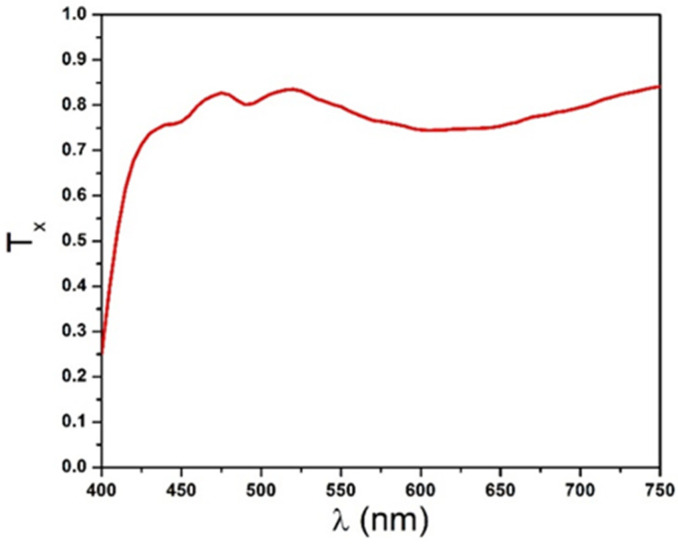
Principal intensity transmittance of the polarizers obtained using Equation (26).

**Figure 7 polymers-14-01805-f007:**
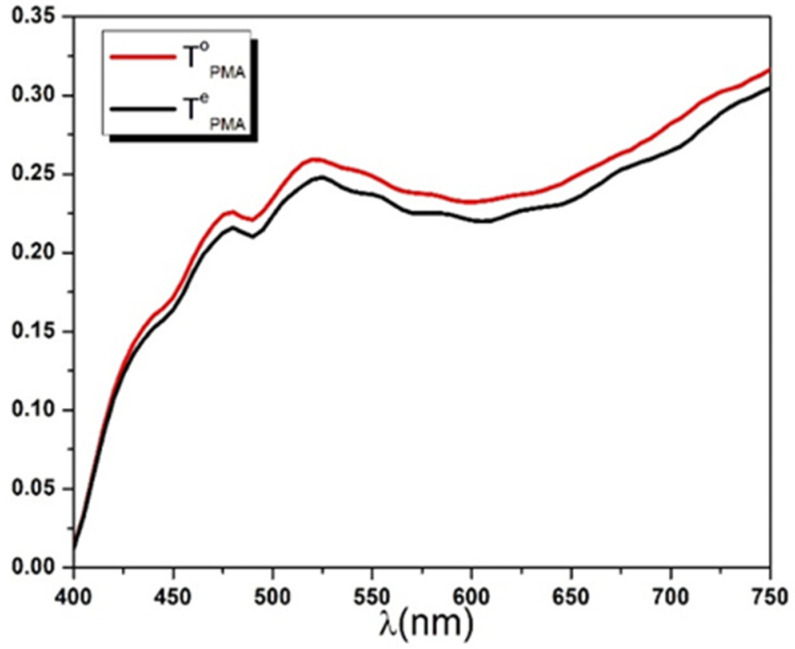
Ordinary and extraordinary intensity transmittances of the Polarizer-Mylar-Analyzer system.

**Figure 8 polymers-14-01805-f008:**
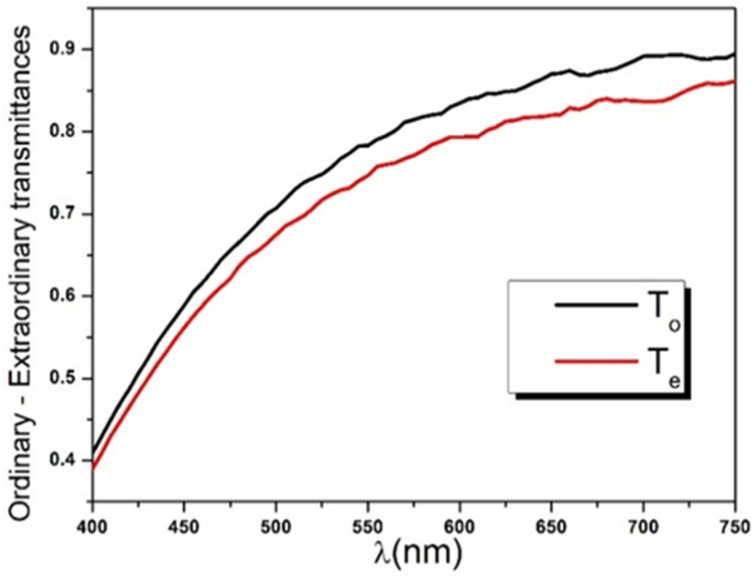
Ordinary and extraordinary intensity transmittances of Mylar sample obtained using Equation (31).

**Figure 9 polymers-14-01805-f009:**
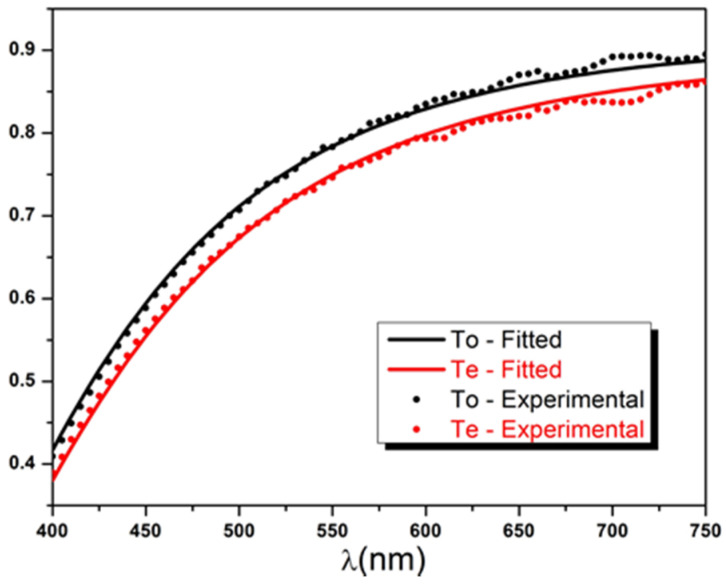
Fitted versus experimental ordinary and extraordinary intensity transmittances of Mylar sample.

**Figure 10 polymers-14-01805-f010:**
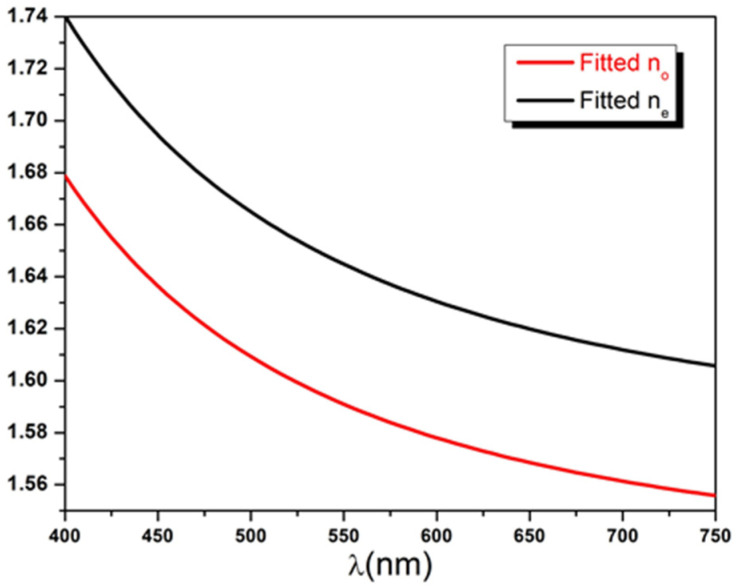
Ordinary and extraordinary refractive indices dispersion curves calculated using Cauchy model and the best fit parameters.

**Figure 11 polymers-14-01805-f011:**
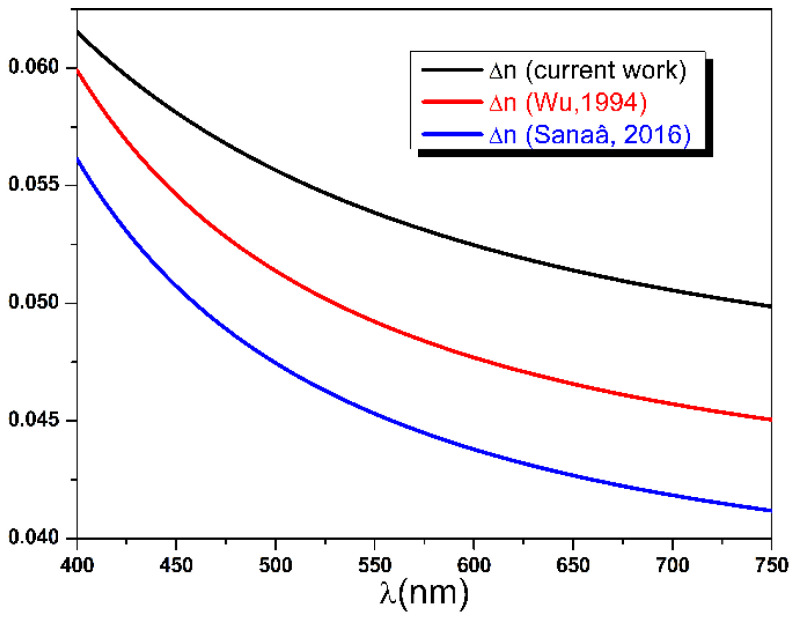
Birefringence dispersion curves in the range $\left(400,\;750\;\mathrm{nm}\right)$ compared with previous experimental results [1,2].

**Figure 12 polymers-14-01805-f012:**
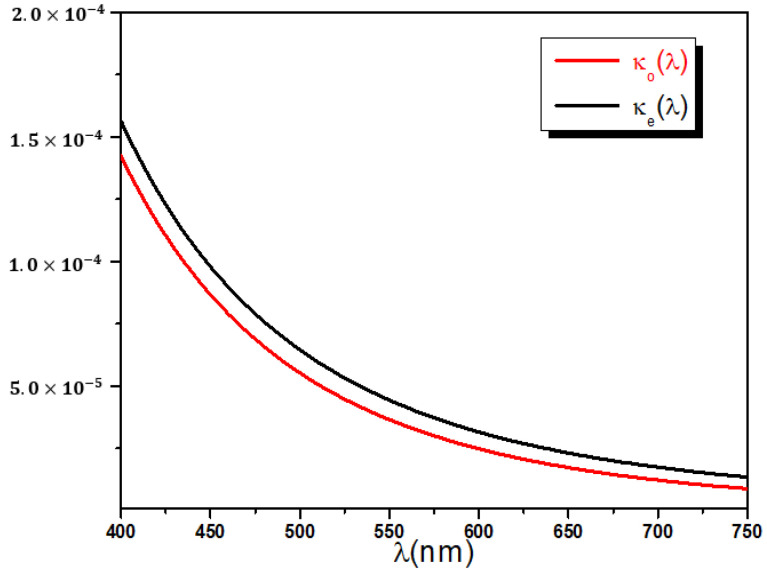
Ordinary and extraordinary extinction coefficients dispersion curves calculated using Cauchy model and the best fit parameters.

**Figure 13 polymers-14-01805-f013:**
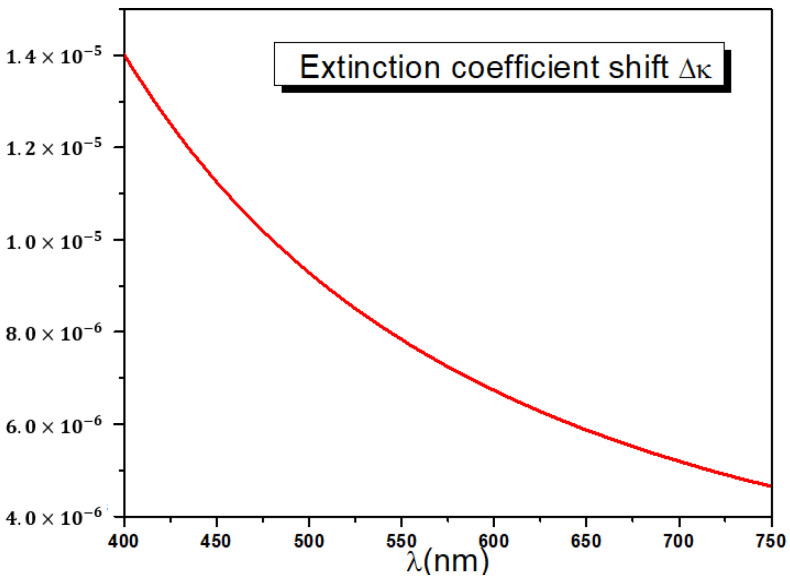
Extinction coefficient shift dispersion curves.

**Figure 14 polymers-14-01805-f014:**
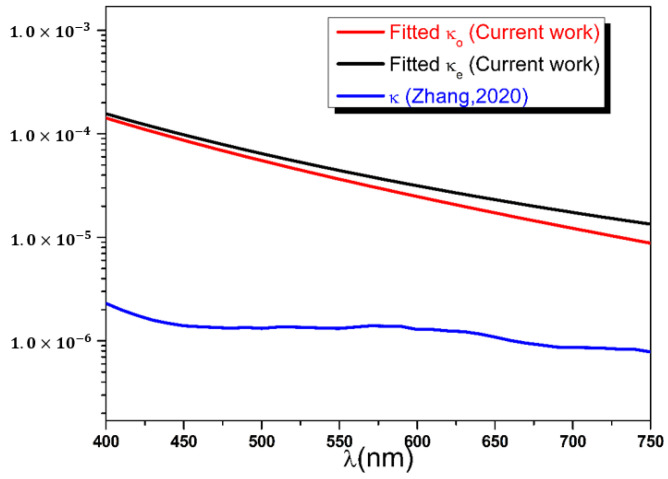
Extinction coefficients’ dispersion curves represented in logarithmic scale and compared with results of reference [11].

**Figure 15 polymers-14-01805-f015:**
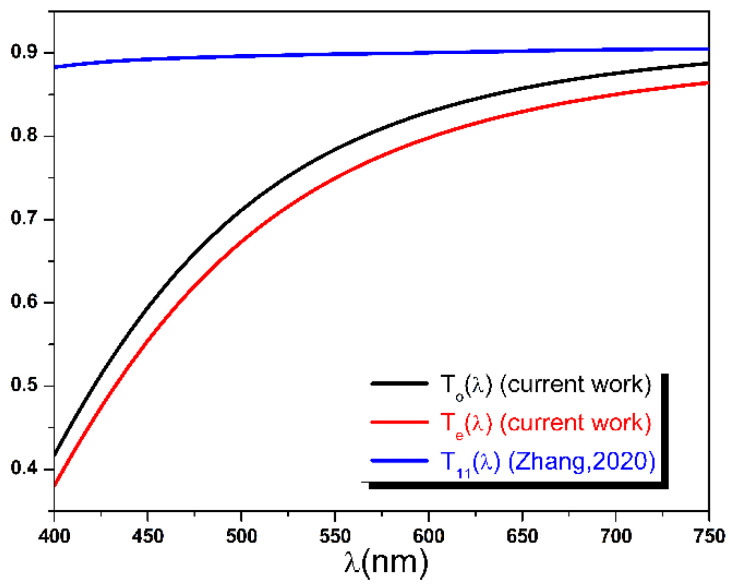
Calculated transmittance using $n\left(\lambda \right)$ and $\kappa \left(\lambda \right)$ of reference [11] compared with fitted $T_{o}\left(\lambda \right)$ and $T_{e}\left(\lambda \right)$ of the current work.

**Table 1 polymers-14-01805-t001:** Fitting parameters of Mylar sheet intensity transmittances $T_{o}\left(\lambda \right)$ and $T_{e}\left(\lambda \right)$.

Best Fit Parameters						
	$A_{n}$	$B_{n}$	$C_{n}$	$A_{\kappa }$	$B_{\kappa }$	$C_{\kappa }$
$T_{o}\left(\lambda \right)$	1.52388	15,249.2	1.52556 × 10^9^	1.322 × 10^−8^	−2.152090	3.989 × 10^6^
$T_{e}\left(\lambda \right)$	1.56911	17,856.4	1.52566 × 10^9^	9.832 × 10^−7^	−0.088029	3.993 × 10^6^
Goodness of fit indicators						
$\chi^{2}$			$R^{2}$		RMSE	
5.13419× 10^−5^			0.999917		6.85588 × 10^−3^	

The units of the Cauchy’s coefficients $B_{n,\kappa }^{o,e}$ and $C_{n,\kappa }^{o,e}$ are ${\mathrm{nm}}^{2}$ and ${\mathrm{nm}}^{4}$, respectively.

## Data Availability

The data presented in this study are available on request from the corresponding author.

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
