# Peer review of "Ordinary and Extraordinary Complex Refractive Indices Extraction of a Mylar Film by Transmission Spectrophotometry"

_polymers, 2022, doi:10.3390/polym14091805_

Round 1
Reviewer 1 Report
The authors creatively proposed a spectrophotometric method to determine both ordinary and extraordinary complex refractive indices (CRIs) of a stretched Polyethylene Terephthalate (Mylar) films, and deduced a new transmittance dispersion formula to describe N and κ wavelength dependence. The proposed theoretical model successfully reproduces the Mylar transmission spectra in the whole visible spectrum range [400-750nm], and its main parameters are comparable to the results in the reported literature. The proposed method has broad application prospects and is recommended to be published.
Modification comments:
It is suggested that the full text should be properly compressed.
Reviewer 2 Report
The paper written by Yassine Makhlouka and coauthors is devoted to a new spectrophotometric technique for the determination of both ordinary and extraordinary complex refractive indices (CRIs) of a stretched Polyethylene Terephthalate (Mylar) film. The paper presents promising results but can be published in Polymer mdpi after major revision. The following issues should be addressed before publication.
- The first remark concerns the "slab approximation" (page 6, line 227). Obviously, the proposed method will be correct only if the optical path length through the specimen satisfies the condition ??>>??, where ?? is the incident light coherence length. It is also obvious that this condition must be satisfied over the entire range of spectral measurements. An experienced reader can guess that the claimed sample thickness of 160 μm fully meets this requirement. However, in my opinion, it would not be superfluous to provide at least an approximate estimate of the coherence length for the light source used. It would also be useful to assess the limits of application of the proposed method in terms of the minimum allowable thickness of the test specimens.
-
Using commercial spectrophotometers, the experimental inaccuracy can lead to missing or wrong solutions especially if the number of independent measurements is equal to the number of unknown quantities. Initial estimates of these quantities are necessary in this case.
The multiwavelength method is based on curve fitting using dispersion equations for n and k. The constants of these equations are varied to obtain the best fit result. Initial estimates of these constants are necessary too.
The construction of the merit function is critical for obtaining accurate results. A weighting factor may be assigned to each measured quantity to take into account the different measurement accuracy.
Unfortunately, there was no place to clarify these issues in the presented work.
- Each new studying method of the optical parameters of dielectric film materials has its advantages and disadvantages and has also its scope where these advantages are most pronounced. Given the film structure of the studied material, the ellipsometric technique is more informative because it involves the simultaneous determination of not only the optical parameters but also the film thickness. Therefore, it is expedient to mention in the Introduction several well-known ellipsometric techniques to more fully substantiate the advantages of the proposed method. In my opinion, the following works (please cite to improve the quality of the manuscript) best characterize the possibilities of ellipsometric techniques in the study of the optical properties of polymers:
https://doi.org/10.1364/AO.54.006208
https://doi.org/10.1063/1.1610236
https://doi.org/10.1016/j.progpolymsci.2014.09.004
Round 2
Reviewer 2 Report
The quality of the paper was essentially improved after revision. Paper can be accepted in its present form.